# TRANSLLAMA: LLM-BASED SIMULTANEOUS TRANSLATION SYSTEM

## ABSTRACT

Decoder-only large language models (LLMs) have recently demonstrated impressive capabilities in text generation and reasoning. Nonetheless, they have limited applications in simultaneous machine translation (SiMT), currently dominated by encoder-decoder transformers. This study demonstrates that, after fine-tuning on a small dataset comprising causally aligned source and target sentence pairs, a pre-trained open-source LLM can control input segmentation directly by generating a special "wait" token. This obviates the need for a separate policy and enables the LLM to perform English-German and English-Russian SiMT tasks with BLEU scores that are comparable to those of specific state-of-the-art baselines. We also evaluated closed-source models such as GPT-4, which displayed encouraging results in performing the SiMT task without prior training (zero-shot), indicating a promising avenue for enhancing future SiMT systems.

## 1 INTRODUCTION

Unlike conventional sequential translation, in which the target text is produced after the end of the corresponding source sentence (or long phrase), in simultaneous machine translation (SiMT) the target text is produced with minimal delay, aiming for the best listener experience expected from professional conference interpreters. While recent years have seen tremendous progress in sentence-based machine translation, mainstream adoption of SiMT systems requires solving a range of technical problems. Perhaps the most important of them is that, much like human conference interpreters, SiMT systems must make optimal decisions about *when* (rather than *how*) to translate. In particular, naively translating each source word immediately results in compromised target quality, given that the meaning of a source word often makes sense only in the context of later words. And while waiting until the end of a sentence might seem a viable solution, in practice it would introduce unacceptable delays between the source and target message. Consequently, the development of an effective SiMT system necessitates striking a balance between these two opposite scenarios.

Existing approaches to maintaining an optimal quality-latency tradeoff in SiMT, conventionally called *policies*, fall into two broad categories: fixed and adaptive. The policy's role is to signal to a separately trained translation model *when* to produce a partial translation (aka WRITE action (Gu et al., 2017)) based of the partial input; at other times the input, which represents either text chunks from an upstream ASR system (in cascade SiMT systems) or speech embeddings (in end-to-end systems), is just read in (READ action). While with a fixed policy (Dalvi et al., 2018; Ma et al., 2019a; Elbayad et al., 2020; Zhang & Feng, 2021), the decision to output translation is based on a simple heuristic, an adaptive policy (Arivazhagan et al., 2019; Ma et al., 2019b; Zhang & Feng, 2022) can be implemented as a separately trained model, for example an agent trained with reinforcement learning (RL) (Gu et al., 2017; Satija & Pineau, 2016).

To the best of our knowledge, state-of-the-art SiMT systems use encoder-decoder transformer architectures in a sequence-to-sequence paradigm. However, as of writing this paper the largest – and generally most expressive – language models are causal decoder-only architectures. We wanted to explore the utility of such models for SiMT tasks, focusing on the English-German and English-Russian language pairs, and specifically if they can be harnessed with minimal engineering effort.

Inspired by the recent success of LLMs – in particular their agential capabilities (Nascimento et al., 2023; Wang et al., 2023a;c) – here we propose TRANSLLAMA, a policy-free SiMT system, in which an off-the-shelf pre-trained decoder-only LLM is fine-tuned on a dataset of causally aligned source

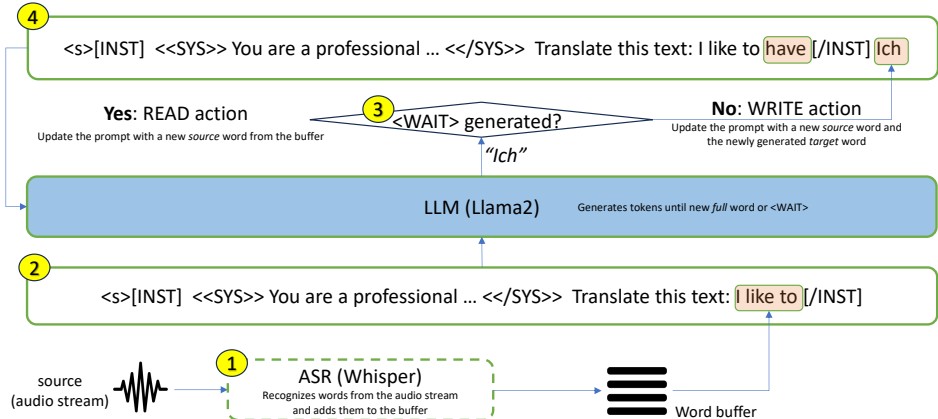

Figure 1: Model overview. The source audio stream is processed with an ASR model (1), which saves each recognized word into the buffer. The initial prompt (2) is built with $k$ source words ($k = 3$ in this example). When the buffer has 3 words, the initial prompt is fed into the LLM, which generates output tokens until either a <WAIT> token or a full word is generated ("Ich" in this example). Then the prompt is updated with a new input ("have") and target ("Ich") word (WRITE action). Finally, the updated prompt (4) is fed back into the LLM. If <WAIT> is generated, the prompt is only updated with a new source word from the buffer (READ action).

and target sentences. The causality of the source is guaranteed by inserting one or more <WAIT> tokens into the target sentence to ensure that target content words never appear earlier than their closest equivalents in the source. We call our model policy-free, because as a result of fine-tuning on a causally aligned dataset the LLM becomes capable of deciding when to output translation and when to read in more of the source, without requiring a separate policy. At inference, the fine-tuned LLM is prompted with *part* of a source sentence concatenated with its corresponding (partial) translation and outputs one or more target tokens until either a full new word or a <WAIT> token is generated, which signals for more words to be read in. When extended with a off-the-shelf ASR model, in addition to text-to-text translation (T2TT), our system handles speech-to-speech translation (S2TT) tasks with quality (as measured by BLEU score (Papineni et al., 2002)) approaching that of some of the recently published baselines at comparable latencies.

Our main contributions are as follows:

1. We present the first system that leverages a decoder-only causal LLM for the SiMT task;

2. We propose a way to fine-tune a pre-trained LLM with direct supervision on a dataset of causally aligned source-target sentence pairs;

3. We demonstrate that an LLM can perform both simultaneous translation and input segmentation without a separate policy, with performance approaching or exceeding state of the art.

The rest of the paper is structured as follows. Section 2 offers a brief overview of most recent SiMT literature. In Section 3 we detail our system's architecture, fine-tuning data preparation and training procedure. In Section 4 we showcase its performance on en-de and en-ru language directions. We conclude with Section 5 in which we discuss the limitations and directions for future work.

## 2 RELATED WORK

SiMT systems aim to deliver the best translation quality, usually measured with BLEU score (Papineni et al., 2002), while keeping its latency at an acceptable level. This quality-latency trade-off is controlled by the "policy", which decides *when* to translate (i.e. perform a WRITE action) and when to receive more input (i.e. perform a READ action). The various policies described in the literature can be broadly categorized into fixed and adaptive (Zhang et al., 2020). Fixed policies (e.g, *wait-k*

(Ma et al., 2019a)) are simple rules that determine the timing and order of WRITE and READ actions irrespective of the context. Early SiMT systems used *chunk-based* approaches (Fügen et al., 2007; Bangalore et al., 2012; Yarmohammadi et al., 2013; Sridhar et al., 2013), in which the input is split into sub-sentence phrases and translated independently of the previous chunk's context, which compromised translation quality. Attempting to overcome this limitation, Dalvi et al. (2018) proposed an *incremental decoding* approach, in which chunk translations incorporate previous context encapsulated by an RNN's hidden states. They showed that coupled with a simple segmentation strategy, their approach outperformed existing state of the art. On the other hand, adaptive policies (e.g. *wait-if* rules (Cho & Esipova, 2016)) make READ/WRITE actions more flexibly by taking account of the partial source and/or target. Adaptive policies can be implemented as separately trained agents (e.g. with reinforcement learning) (Grissom II et al., 2014; Gu et al., 2017; Satija & Pineau, 2016; Alinejad et al., 2018). In such policies, READ/WRITE actions can be taken based on attention Raffel et al. (2017); Chiu & Raffel (2018); Arivazhagan et al. (2019); Ma et al. (2020b), or stability of the model's outputs over $n$ steps (so-called *local agreement* (Liu et al., 2020a; Ko et al., 2023; Polák et al., 2022)). More recent studies have also explored training the policy with binary search Guo et al. (2023) aiming to maximize the gain in translation quality per each token read, or cast the problem of deciding when to translate as a hidden Markov transformer Zhang & Feng (2023), in which hidden events correspond to the times at which to output translation.

Another promising line of work, related to the present study, aims to fine-tune encoder-decoder transformers, such as mBART (Liu et al., 2020b), originally pre-trained for sentence-level translation, for the SiMT task. For example, Fukuda et al. (2023); Kano et al. (2022) utilized fine-tuning on prefix-alignment data and Zhang et al. (2020) on meaningful units, achieving compelling performance on some language pairs.

Distinct from these approaches, we propose to fine-tune a large langauge model for the SiMT task on a dataset of causally aligned source-target sentence pairs, which we describe below.

## 3 METHOD

Although the LLMs we consider in this paper are designed to process only text input, we add an ASR stage to enable it to also perform S2TT mode. Thus, we follow a cascaded approach shown in Fig. 1.

**Causal alignment.** Training SiMT models, including optimal segmentation policies, with direct supervision has remained a challenge (Guo et al., 2023) due to at least three reasons: (1) word order inconsistencies between the source and target, (2) omissions of words from the target that were present in the source, and/or (3) additions of words to the target not explicitly present in the source, making it difficult to establish unambiguous correspondences between each source and target words. This is less of a problem for offline translation models, because they are trained with direct supervision on pairs of *complete* source and target sentences, and both during training and inference the entire source context is revealed. However, it is not immediately clear how to use direct supervision for the SiMT task, in which the model must begin translation based on *partial* context. Nevertheless, we believe that direct supervision for the SiMT task is possible and propose a way to accomplish that with a *causally aligned* dataset. In such a dataset, a target word never appears before its corresponding (when such correspondence can be established) source word in time, which is defined as the number of words from the sentence start. In other words, in a causally aligned source-target sentence pair, source words are guaranteed to be causal relative to their corresponding target words. We illustrate this in Fig. 2.

Note that the causal alignment is not always perfect: due to the word length mismatch between the source and target, not all all source words will have a corresponding target word, and vice versa, not every target word will have a corresponding word in the source. However, as we demonstrate below, fine-tuning an LLM on such a causally aligned dataset enables it to achieve results comparable to some state-of-the-art baselines.

In order to causally align the source and target, we split each sentence using the `word_tokenize` function from the *nltk* package (Bird et al., 2009), treating punctuation marks as "words", then find the best correspondences between the source and target words with *SimAlign* (Jalili Sabet et al., 2020), and finally insert as many `<WAIT>` tokens into the target as appropriate. If after alignment

| | original | | | causally aligned | | | | original | | | causally aligned | |
|---|---|---|---|---|---|---|---|---|---|---|---|---|
| | **en** | **ru** | | **en** | **ru** | | | **en** | **de** | | **en** | **de** |
| 1 | They → Они | | | They → Они | | | 1 | He → Er | | | He → Er | |
| 2 | live → живут | | | live → живут | | | 2 | took befreite | | | took @ | |
| 3 | in глубоко | | | in @ | | | 3 | one uns | | | one @ | |
| 4 | the в | | | the @ | | | 4 | of von | | | of @ | |
| 5 | depths конголезских | | | depths → глубоко | | | 5 | the einer | | | the @ | |
| 6 | of джунглях | | | of в | | | 6 | worst der | | | worst @ | |
| 7 | the , | | | the @ | | | 7 | scourges schlimmsten | | | scourges @ | |
| 8 | Congolese где | | | Congolese → конголезских | | | 8 | of Geißeln | | | of @ | |
| 9 | jungle сложно | | | jungle → джунглях | | | 9 | mankind der | | | mankind @ | |
| 10 | and проводить | | | and , | | | 10 | away Menschheit | | | away befreite | |
| 11 | it исследования | | | it где | | | 11 | from . | | | from @ | |
| 12 | has . | | | has @ | | | 12 | us | | | us → uns | |
| 13 | been | | | been @ | | | 13 | . | | | . von | |
| 14 | very | | | very @ | | | 14 | | | | _ _ einer | |
| 15 | difficult | | | difficult → сложно | | | 15 | | | | _ _ der | |
| 16 | to | | | to проводить | | | 16 | | | | _ _ schlimmsten | |
| 17 | study | | | study → исследования | | | 17 | | | | _ _ Geißeln | |
| 18 | them | | | them @ | | | 18 | | | | _ _ der | |
| 19 | . | | | . → . | | | 19 | | | | _ _ Menschheit | |
| | | | | | | | 20 | | | | _ _ . | |

Figure 2: **Causality-preserving alignment.** Two examples are shown: for en-ru (left) and en-de (right). If time is defined as the number of words from the beginning of the sentence, before alignment, some target words appear earlier than their corresponding English equivalents in the source. By inserting <WAIT> tokens (shown as "@"), we can shift those target words into the future, thereby achieving causality for every content word. "_ _" are fillers added at the end of the source sentence if neccesary to match its length with that of the target.

the target becomes longer than the source due to added <WAIT> tokens, we pad the source at the end with filler strings ensuring that the aligned source and target sentences have the same number of "words". These filler strings are only used for convenient batching and are dropped before tokenization.

**Supervised Fine-Tuning.** We fine-tune the LLAMA-2 13B and and 70B models (Touvron et al., 2023) [1] to optimize the following objective:

$$\mathcal{L}_{\text{T2TT}} = -\sum_{t=1}^{|y|} \log p(y_t | y_{<t}, x_{\leq t}) \tag{1}$$

where $y_t$ is the next target token, $y_{<t}$ are previously generated (and committed) tokens and $x_{\leq t}$ and the partial source tokens revealed up to the time step $t$. Following (Touvron et al., 2023), we zero out the loss on tokens corresponding the to system message and source, only backpropagating on the target tokens.

We use batches of prompt-response pairs collated in the following way. Before tokenization, each aligned sentence-target pair selected from the causally aligned dataset is trimmed from the right to leave first $l$ words, where $l \sim U(1, L)$ and $L$ is the full length of the causally aligned source-target pair. After trimming, all the <WAIT> tokens except the last one (if present) are dropped, because they are never plugged back into the input and only serve the purpose of signaling for more words

---

[1] We found that the LLAMA-2-CHAT variants (both 13B and 70B), when fine-tuned on our causally aligned dataset performed slightly, but consistently, worse than LLAMA-2, and we report the results for the latter model only.

to be read in. Likewise, we drop all the fillers (if present) from the source. Finally, the system message, trimmed source and trimmed target are joined into the prompt (as shown in Fig. 4) and tokenized. Because there is no <WAIT> token in the LLAMA 2 tokenizer, we use 0 (which originally corresponds to the <UNK> token). Thus, the model is fine-tuned to either output the next token of a word or <WAIT>, if the partial source does not contain sufficient information needed to predict translation.

To save memory, we loaded the the base model in 4-bit precision. This allowed us to fine-tune LLAMA 2 70B on one NVIDIA A100 80GB device. We fine-tune the base model with LoRA (Hu et al., 2022) with $r = 16$ and $\alpha = 32$ for 3 epochs with a batch size of 25 and gradient accumulation of 4 steps. We save model checkpoints every 10 steps and select the one with the lowest validation loss for inference. For optimization, we used the paged_adamw_32bit optimizer with default parameters, and a learning rate schedule with a linear warm-up of 10 steps up to 0.00005, followed by a cosine decay. For parameter-efficient training, as well as for inference, we used the transformers[2] library.

**Inference.** At inference, given a prompt (Fig. 4) comprised of a system message, partial source and previously committed partial target, the LLM greedily generates one or more next tokens. We use modified wait-*k* (Ma et al., 2019a), in which WRITE actions are only allowed when the length of the PARTIAL_SOURCE is equal or greater than $k$. Since $k$ controls the tradeoff between quality and latency, we report results for different values of $k$. After a full new word – which may consist of several tokens – is generated, the prompt is updated by appending a new source word to the partial source and the newly generated word to the partial target. This process is repeated until the LLM generates the <EOS> token. All the generation parameters were at default, except top_p which we set to 0.7. We did not use beam search during generation.

| PARTIAL_SOURCE | PARTIAL_TARGET | Prediction |
|---|---|---|
| I | | <WAIT> |
| I like | | Я |
| I like to | Я | люблю |
| I like to have | Я люблю | <WAIT> |
| I like to have tea | Я люблю | пить |
| I like to have tea | Я люблю пить | чай |
| I like to have tea in the | Я люблю пить чай | <WAIT> |
| I like to have tea in the morning. | Я люблю пить чай | по |
| I like to have tea in the morning. | Я люблю пить чай по | утрам. |
| I like to have tea in the morning. | Я люблю пить чай по утрам. | <EOS> |

Figure 3: An illustration of the inference process for the en-ru language pair. Assuming $k = 1$, given the prompt with one source and zero target words, the model first outputs <WAIT>, which signals for the next source word to be read in. At the next step, the model generates the first target word (Я), which is plugged into the prompt at the next step. This process continues until <EOS> is generated.

After all the source words have been revealed, the input is no longer partial and no new words are added to it, but the generation process continues until <EOS>. Importantly, if the model generates the <WAIT> token, a new source word is read in, but the <WAIT> token itself is not appended to the partial target. We illustrate the inference process in Fig. 3 and Algorithm 1.

**Prompt structure.** We follow a similar prompt structure as in Touvron et al. (2023) (Fig. 4). For the SYSTEM_MESSAGE we used the following text: *"You are a professional conference interpreter. Given an English text you translate it into* {TARGET_LANGUAGE} *as accurately and as concisely as possible, NEVER adding comments of your own. You output translation when the information available in the source is unambiguous, otherwise you output the wait token (*{WAIT_TOKEN}*), not flanked by anything else. It's important that you get this right."*. We note that while the system message is only necessary in zero-shot SiMT scenarios – which we discuss below – for consistency we still kept it in all the experiments reported here, including those involving supervised fine-tuning.

---

[2]https://huggingface.co/docs/transformers/installation

---

**Algorithm 1** Inference process

---

```
partial_target = []
k = WAIT_K

while True:
    partial_source = SOURCE[:k]
    prompt = " ".join([SYS_MSG, partial_source, partial_target])

    # generate until next full word, or <EOS> or <WAIT>
    next_word = model.generate(prompt)

    if next_word == "<EOS>":
        break                                 # finish sentence
    elif next_word == "<WAIT>":
        k += 1                                 # READ action
    else:
        partial_target.append(next_word)  # WRITE action
        k += 1
```

---

```
[INST]
<<SYS>>

[SYSTEM_MESSAGE]
<</SYS>>
Translate this text: PARTIAL_SOURCE [/INST] PARTIAL_TARGET
```

Figure 4: Prompt structure.

**Automatic speech recognition**. Given that the LLMs are designed to process text input, to enable S2TT we first need to extract text from input audio, for which we use Whisper [3] (Radford et al., 2023). Specifically, for each READ action, a new segment of audio, lasting 200 ms, is added to any previously read audio chunks and then processed by Whisper. This method of fixed audio windowing often results in partially clipped words. To address this, we discard the last word predicted by Whisper during each READ action unless the entire source audio has been read in. We note that this approach to online ASR is somewhat naive and has room for improvement – as indicated by a roughly 1 BLEU point decrease due to ASR-related errors (Fig. 9). Since our main objective was to assess the capability of LLMs to perform SiMT tasks, we leave exploring ways to decrease ASR errors to future work.

## 4 RESULTS

**Data.** For supervised fine-tuning (SFT), validation and testing, we used MuST-C v2.0 (Di Gangi et al., 2019) for English-to-German (en-de) and English-to-Russian (en-ru) translation direction. We randomly selected 4000 sentences for training and 100 sentences for validation. However, since it is possible that the dataset that LLAMA2 was pre-trained on and MuST-C v2.0 (including its validation and test set) might have overlapping content, we also compiled another test set, which we call NEW-TED-2023. This test set has a similar content type (TED talks) and follows the same format as the original MuST-C v2.0, but only includes talks posted after February 2023. The dataset has two parts: 102 source-target pairs for en-de and 102 for en-ru language pair. Unless indicated otherwise, we report the results obtained on this test set.

**T2TT**. We first analyzed the T2TT performance or our approach on the MuST-C dataset v2.0 (Di Gangi et al., 2019). To get a sense for the quality-latency tradeoff, we plot BLEU scores against several different values of $k$ (because $k$ is the only way to control the translation latency). The

---

[3] We used whisper-large-v2.

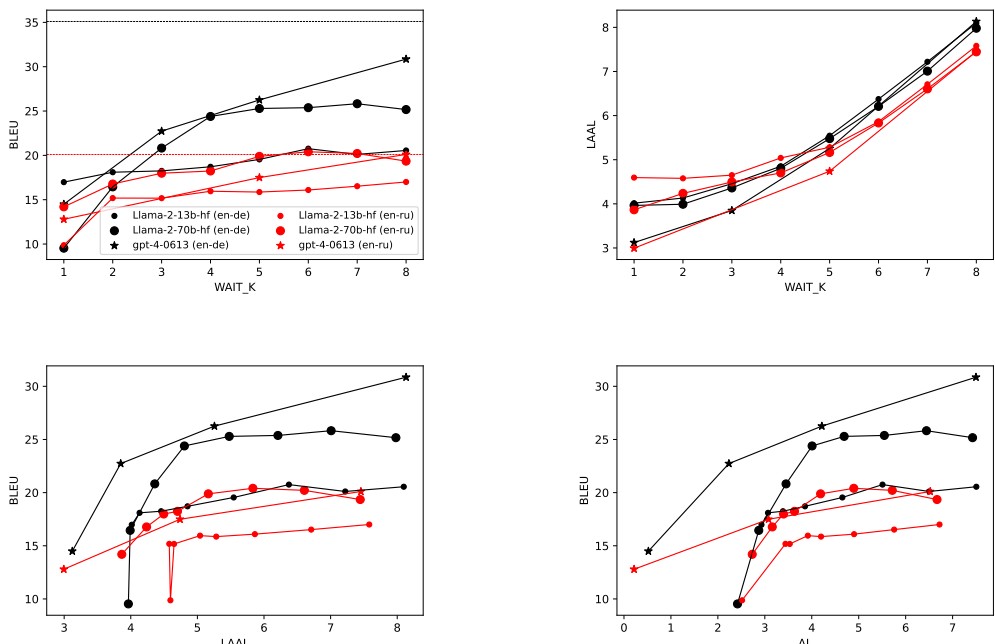

Figure 5: Dependence of latency and quality on $k$ (top panels) and quality-latency tradeoff curves (bottom panels) for the T2TT mode on the MuST-C v2.0 dataset. For reference, dashed lines indicated GPT-4's sentence-level (i.e. with $k$ set to the sentence length) BLEU scores: black for `en-de` and red for `en-ru`.

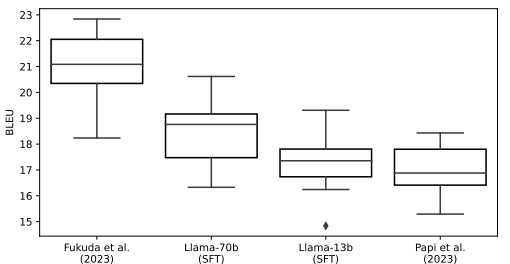

Figure 6: S2TT performance of SFT LLAMA-2 and two recently published models on the `en-de` language pair on TED-TST-2023. See also Appendix C.1.

Figure 7: Zero-shot S2TT performance or our approach compared with GPT-3.5 and GPT-4 on the `en-de` language pair on TED-TST-2023.

results, shown in Fig. 5, suggest that the LLM's size is a major factor determining the translation quality.

**S2TT**. We next test fine-tuned LLMs and compare them with two recently published S2TT baselines (Fukuda et al., 2023; Papi et al., 2023) as well as to OpenAI's GPT-3.5 and GPT-4 (in zero-shot mode). To ensure as fair a comparison as possible, we ensured that average lagging (AL) of all of the models below approximately 2000 ms. For Llama-2 models we set $k = 5$ (the other models' settings are listed in Appendix D). The boxplots in Figs. 6, 7 and throughout are drawn based on data from 10 evaluation runs of the same model with the same parameters on sentence pairs sampled with replacement from TED-TST-2023. The results show a degradation of translation quality by approximately 1 BLEU score point compared to T2TT mode, which is to be expected due to ASR errors (Fig. 9).

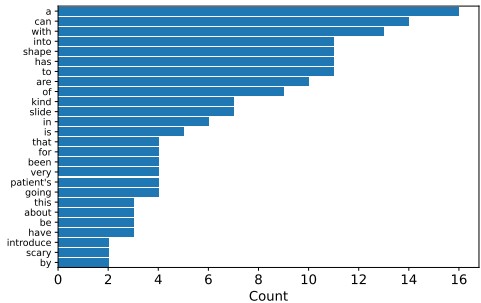

Figure 8: After fine-tuning, LLAMA-2 generates `<WAIT>` tokens predominantly after function words (especially articles and prepositions).

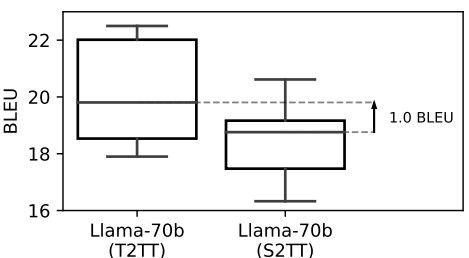

Figure 9: Performance decrease due to ASR-related errors. In T2TT mode, `Llama2-70b` performs by about 1 BLEU score point better than the the same model on the same data in S2TT mode.

| $k$ | w/ `<WAIT>` | w/o `<WAIT>` |
|---|---|---|
| 1 | 15.23 | 14.88 |
| 2 | 17.17 | 15.66 |

(a)

| $k$ | w/ `<WAIT>` | w/o `<WAIT>` |
|---|---|---|
| 1 | 14.76 | 10.80 |
| 2 | 14.97 | 11.94 |
| 4 | 17.42 | 15.67 |

(b)

| $k$ | w/ `<WAIT>` | w/o `<WAIT>` |
|---|---|---|
| 1 | 17.17 | 4.64 |
| 2 | 16.83 | 7.84 |
| 4 | 19.24 | 14.80 |

(c)

Table 1: Removing the instruction to generate or suppressing the `<WAIT>` token degrades performance. The numbers indicate BLEU scores on TED-TST-2023 (`en-de`) in T2TT mode for GPT-4 (a), supervised fine-tuned Llama-2-13b-hf (b) and Llama-2-70b-hf (c).

**Zero-shot T2TT.** Can the LLMs perform the SiMT task zero-shot, that is without any prior fine-tuning? To answer this question, we used LLMs that have been fine-tuned with RLHF for instruction following: open-source LLAMA2-CHAT, as well as GPT-3.5 (`gpt-3.5-turbo-0613`) and GPT-4 (`gpt-4-0613`), which were among the strongest closed-source LLMs available at the time of writing this paper. In general, with the notable exception of GPT-4, zero-shot performance was poor. Inspection of the translations revealed that the models consistently failed to follow the prompt instruction, specifically, (1) generating output in English rather than the target language, (2) adding expressly prohibited explanatory comments, (3) restating or summarizing the task, or (4) explaining the reason for adding `<WAIT>` tokens). GPT-4 was surprisingly good, performing better than the supervised fine-tuned LLAMA2-70B, and we speculate that the performance of GPT-3.5 and GPT-4 could be further improved with SFT [4], more sophisticated generation strategies and prompt engineering.

**Importance of wait tokens.** To evaluate the utility of `<WAIT>` tokens, we conduct two ablation experiments. In the first experiment we consider a zero-shot translation scenario in which GPT-4 was not instructed to use `<WAIT>` tokens. In the second experiment, we suppress the generation of `<WAIT>` tokens in supervised fine-tuned LLMs. The results, as indicated in Table 1, reveal that GPT-4 demonstrates marginally inferior performance when $k \in \{1, 2\}$[5] when not instructed about `<WAIT>` tokens. However, it is important to note that in a zero-shot context, the GPT-3.5 and GPT-4 seldom generated `<WAIT>` tokens (almost never for $k > 2$). Therefore, the directive to employ these tokens only exhibited a discernible impact for smaller values of $k$. By contrast, in the SFT scenario, suppressing `<WAIT>` tokens led to significantly decreased performance for both the 13 and 70B versions of LLAMA-2 (Table 1 (b, c)).

To gain insight into where LLAMA-2 tended to insert the `<WAIT>` token, we plot the distribution of words after which the SFT models generated this token. Fig. 8 shows that most of the time the model generated `<WAIT>` after function words – which makes sense – rather than content words, indicating that it had learned to choose appropriately between READ and WRITE actions.

---

[4]SFT was not available for GPT-3.5 and GPT-4 at the time of writing this paper.

[5]We did not investigate the role of `<WAIT>` tokens for $k > 2$, because GPT-4 almost never generates them for those values of $k$.

## 5 CONCLUSION AND FUTURE DIRECTIONS

We have shown that with minimal fine-tuning and without resorting to sophisticated training techniques (e.g. checkpoint averaging (Fukuda et al., 2023)), an off-the-shelf pre-trained LLM can perform simultaneous translation and achieve encouraging results that rival some of the recent SiMT models. This opens interesting directions to be explored in future work, such as multilingual fine-tuning, self-instruct (Wang et al., 2023b) and human preference tuning (Ouyang et al., 2022).

There are several reasons to believe that we are far from unlocking the full potential of LLMs for SiMT. First, we followed the practice – standard in the SiMT literature – of evaluating the model on individual sentences randomly sampled from continuous prose. However, many (if not the majority of) short sentences are ambiguous when taken out of context. Even human conference interpreters routinely prepare for an upcoming translation job, studying relevant materials, which means that they do not have to translate sentences taken out of context. For this reason, we believe that the most straightforward way to boost the performance of future LLM-based SiMT systems is to insert background information into the prompt. Second, the big difference in zero-shot performance between GPT-3.5 and GPT-4 suggests that size is likely the biggest factor determining the model's translation quality, and that further gains can be achieved once SFT becomes available for these closed-source models.

In conclusion, we note that there are several performance bottlenecks that must be addressed before our approach can be deployed for simultaneous translation in the real world. As we show in Appendix E, these bottlenecks result from a long system message, which is often longer than the source sentence itself, as well as delays introduced by the ASR sybsystem and weight quantization. We believe that these issues are not prohibitive. Specifically, instead of using a separate ASR model, future work might follow an end-to-end approach similar to Fathullah et al. (2023), in which instead of being converted into text with an separate ASR model, the audio is directly mapped into the LLM's embedding space, reducing the system's overall latency. Efficient quantization schemes, faster algorithms and hardware support for low bit-width arithmetic are also promising directions. Finally, because LLAMA-2 was trained predominantly on English text, its tokenizer represents English more efficiently than other languages. That is, fewer tokens on average are needed to encode a text in English than a text of the same length (in characters) in another, less represented, language. Thus, future LLMs pre-trained on a linguistically more balanced dataset, might be slightly faster at inference.

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

## A    SUPPLEMENTARY RESULTS FOR THE S2TT TASK

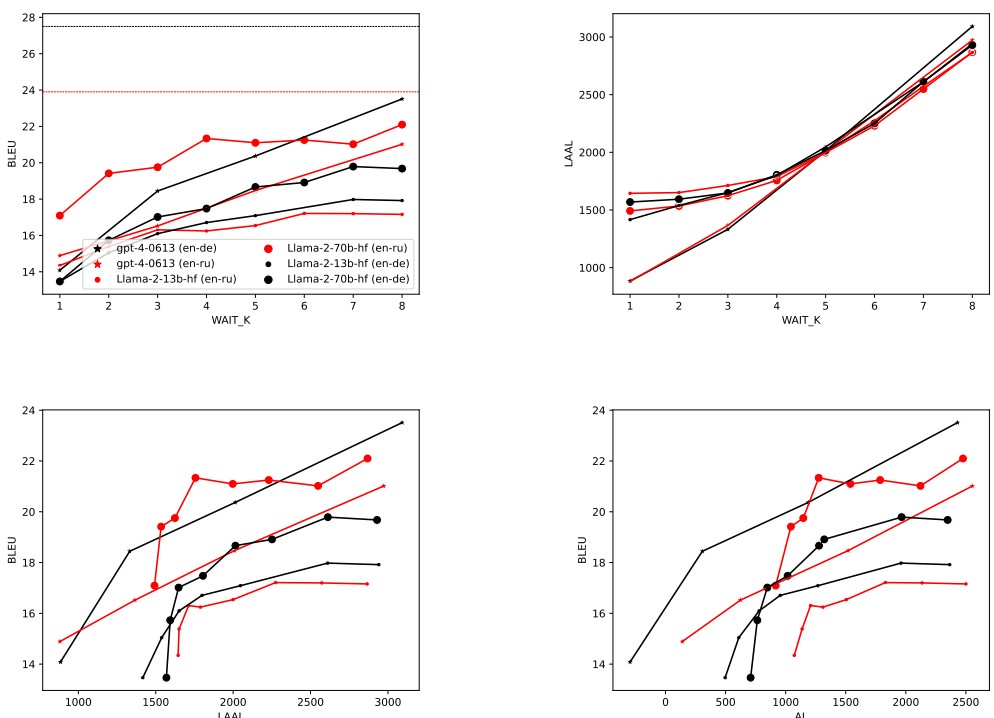

Figure 10: Dependence of latency and quality on $k$ (top panels) and quality-latency tradeoff curves (bottom panels) for the S2TT mode on the NEW-TED-2023 dataset. For reference, dashed lines indicated GPT-4's sentence-level (i.e. with $k$ set to the sentence length) BLEU scores: black for `en-de` and red for `en-ru`.

## B    ENGLISH-RUSSIAN S2TT TASK ($k = 5$)

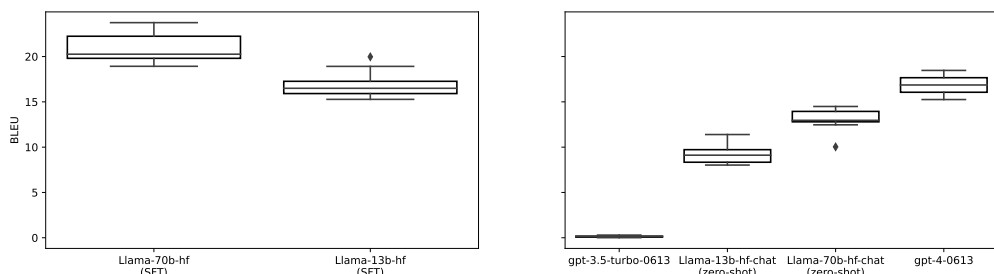

Figure 11: S2TT `en-ru` performance of our method on TED-TST-2023. Left panel: supervised fine-tuned LLAMA-2. Right panel: zero-shot S2TT performance of LLAMA-2-CHAT. All the runs were on TED-TST-2023, with $k = 5$ to ensure AL around 2000 ms. Each of the boxplots is drawn based on data from 10 evaluation runs on sentences randomly sampled with replacement from the test set.

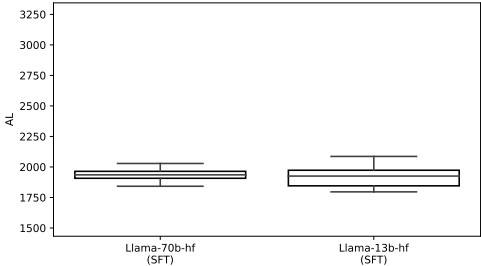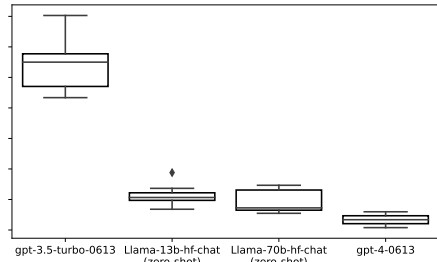

Figure 12: Average lagging in S2TT mode for the English-Russian language pair. Left panel: supervised fine-tuned LLAMA-2. Right panel: zero-shot S2TT performance of LLAMA-2-CHAT. All the runs were on TED-TST-2023, with $k = 5$ to ensure AL around 2000 ms. Each of the boxplots is drawn based on data from 10 evaluation runs on sentences randomly sampled with replacement from the test set.

## C  ADDITIONAL PERFORMANCE DATA FOR THE S2TT TASK

### C.1  ENGLISH-GERMAN

Here we report additional comparisons including latency performance measured using several different metrics, including Average Lagging (AL) (Ma et al., 2019a), Length Adaptive Average Lagging (LAAL) (Papi et al., 2022), Average Proportion (AP) (Cho & Esipova, 2016) and Differentiable Average Lagging (DAL) (Cherry & Foster, 2019).

| System | BLEU | LAAL | AL | AP | DAL |
|---|---|---|---|---|---|
| gpt-3.5-turbo-0613 (zero-shot) | 2.08 (0.24) | 2637.11 (252.79) | 2574.98 (230.95) | 0.35 (0.0) | 2477.55 (146.26) |
| gpt-4-0613 (zero-shot) | 21.82 (2.81) | 2448.86 (74.74) | 1998.63 (110.91) | 0.94 (0.03) | 2813.47 (69.48) |
| Llama-70b-hf (SFT) | 18.41 (1.4) | 2107.57 (59.68) | 1619.64 (76.47) | 0.84 (0.02) | 2454.72 (67.84) |
| Llama-13b-hf (SFT) | 17.07 (0.68) | 2358.89 (34.11) | 1880.76 (61.77) | 0.88 (0.02) | 2735.34 (40.88) |
| Papi et al. (2023) | 17.01 (1.0) | 2295.72 (41.54) | 1867.1 (148.69) | 0.77 (0.01) | 3251.38 (139.12) |
| Fukuda et al. (2023) | 21.08 (1.41) | 2005.39 (71.04) | 1397.33 (85.74) | 0.9 (0.01) | 3066.15 (122.01) |

Table 2: Mean performance metrics of Llama-2 (SFT) compared to some recent S2TT systems and GPT-3.5 and GPT-4 (zero-shot). Then mean and standard deviation (in brackets) are computed over 10 runs of the same model on 102 source-target pairs sampled with replacement from TED-TST-2023.

### C.2  ENGLISH-RUSSIAN

| System | BLEU | LAAL | AL | AP | DAL |
|---|---|---|---|---|---|
| gpt-3.5-turbo-0613 (zero-shot) | 0.14 (0.1) | 2876.85 (240.03) | 2861.22 (245.91) | 0.28 (0.04) | 2661.22 (231.0) |
| gpt-4-0613 (zero-shot) | 16.86 (2.27) | 2022.81 (20.3) | 1584.38 (91.81) | 0.82 (0.04) | 2390.11 (23.65) |
| Llama-70b-hf (SFT) | 20.96 (1.71) | 2252.75 (49.77) | 1937.76 (62.75) | 0.9 (0.08) | 2676.56 (62.11) |
| Llama-13b-hf (SFT) | 16.9 (1.52) | 2238.6 (48.38) | 1917.46 (90.38) | 0.87 (0.03) | 2641.01 (45.73) |

Table 3: Mean performance metrics of Llama-2 (SFT) compared to some recent S2TT systems and GPT-3.5 and GPT-4 (zero-shot). Then mean and standard deviation (in brackets) are computed over 10 runs of the same model on 102 source-target pairs sampled with replacement from TED-TST-2023.

# D  PARAMETERS USED FOR COMPARISONS WITH BASELINES ON THE S2ST EN-DE TASK

Papi et al. (2023)

We used the open-source implementation of the model[6]. The evaluations were run in *SimulEval*[7] (Ma et al., 2020a) with the following parameters:

```
extract-attn-from-layer 5
frame-num 2
attn-threshold 0.25
speech-segment-factor 8
```

Fukuda et al. (2023)

The source code for the model and weights were obtained on request from the authors. The evaluations were run in *SimulEval* with the following parameters:

```
source-segment-size 950
la-n 2
beam 5
sacrebleu-tokenizer 13a
```

We chose these parameters aiming to maximize the BLEU score while keeping AL approximately below 2000 ms.

---

[6]https://github.com/hlt-mt/FBK-fairseq/tree/master/fbk_works
[7]https://github.com/facebookresearch/SimulEval

## E  INFERENCE WALL TIME COMPARISONS

Here we compare real-time factors of our model in different sizes and compare them with those of the selected baselines and GPT-4. Real-time factor is the ratio of the amount of time taken to process source audio to the length of the source audio itself [8]. We note that removing the system message from the prompt speeds up inference with no noticeable drop in quality for supervised fine-tuned models. Loading our model's weights with 16-bit (instead of 4-bit) quantization further accelerates inference. Finally, we observe that the use of ASR in S2TT mode substantially reduces system speed. An end-to-end implementation, directly converting raw source audio into the LLM's embedding space, could potentially alleviate this performance bottleneck.

| model | mode | quantization | system message | size, bn param. | RTF |
|---|---|---|---|---|---|
| Ours | T2TT | 16-bit | no | 13 | 1.7 |
| Ours | T2TT | 4-bit | no | 13 | 2.2 |
| Ours | T2TT | 16-bit | yes | 13 | 2.9 |
| Ours | T2TT | 4-bit | yes | 13 | 4.2 |
| Ours | S2TT | 16-bit | no | 13 | 5.9 |
| Ours | S2TT | 4-bit | no | 13 | 7.6 |
| Ours | S2TT | 16-bit | yes | 13 | 8.0 |
| Ours | S2TT | 4-bit | yes | 13 | 9.3 |
| Ours | T2TT | 4-bit | no | 70 | 14.6 |
| Ours | T2TT | 4-bit | yes | 70 | 20.2 |
| Ours | S2TT | 4-bit | no | 70 | 15.3 |
| Ours | S2TT | 4-bit | yes | 70 | 23.9 |
| GPT-4 | T2TT | unknown | yes | unknown | 1.5 |
| GPT-4 | S2TT | unknown | yes | unknown | 4.8 |
| Fukuda et al. (2023) | S2TT | 16-bit | N/A | 1.04 | 0.7 |
| Papi et al. (2023) | S2TT | 16-bit | N/A | 0.176 | 1.4 |

Table 4: Inference wall time comparisons for our model and selected baselines on `en-de`. Real-time factor (RTF) less than one means the model is faster than real time.

---

[8]For T2TT mode, the source audio transcripts were fed directly in the LLM. The RTF was calculated based on the known length of the audio corresponding to the source transcripts and the time to complete translation of that text.

## F  EXAMPLES OF TRANSLATION

en-de

Source: But in Antarctica, some scientists who are normally pretty levelheaded are getting a little bit freaked out, I would say, is a fair definition, about the lowest level ever, at this point in the year, of sea ice.

TransLLaMa: Aber in der Antarktis, einige Wissenschaftler, die normalerweise recht nüchtern sind, werden ein wenig verängstigt, würde ich sagen, ist eine gerechte Definition über den niedrigsten Stand, den wir jemals zu diesem Zeitpunkt im Jahr von Meereis hatten.

Back translation: But in Antarctica, some scientists who are normally quite sober are getting a little frightened, I would say, is a fair definition of the lowest level we've ever had of sea ice at this point in the year.

Source: I remember when there were so many fossil fuel delegates in Madrid, but by the time we got to Glasgow a year and a half ago, the delegates from the fossil fuel companies made up a larger group than the largest national delegation.

TransLLaMa: Ich erinnere mich an die Zeit, als es so viele Delegierte der fossilen Brennstoffind-ustrie in Madrid gab, aber als wir ein Jahr und ein halbes später in Glasgow waren, machten die Delegierten der fossilen Brennstoffunternehmen eine größere Gruppe aus als die größte nationale Delegation.

Back translation: I remember when there were so many fossil fuel industry delegates in Madrid, but when we were in Glasgow a year and a half later, the fossil fuel company delegates made up a larger group than the largest national delegation.

Source: And I'm a really positive person, so it's actually difficult to say this, but the way we do philanthropy right now, the way we've done it for decades, is broken.

TransLLaMa: Ich bin eine sehr positive Person, also ist es für mich eigentlich schwierig, das zu sagen, aber die Art und Weise, wie wir Philanthropie betreiben, die Art und Weise, wie wir es seit Jahrzehnten betreiben, ist kaputt.

Back translation: I'm a very positive person, so it's actually hard for me to say this, but the way we do philanthropy, the way we've done it for decades, is broken.

Source: Work norms, of course, differ around the world, but we replicated our original American result in South Korea, which is known by the numbers to be one of the hardest-working countries in the OECD, and in France, which is known for other strengths.

TransLLaMa: Natürlich unterscheiden sich die Arbeitsnormen in der Welt, aber wir konnten unser ursprüngliches Ergebnis in den USA in Südkorea wiederholen, das bekannt ist als eines der härtesten arbeitenden Länder in der OECD, und in Frankreich, das für andere Stärken bekannt ist.

Back translation: Of course, labor standards vary around the world, but we were able to replicate our original result in the US in South Korea, known as one of the hardest working countries in the OECD, and in France, known for other strengths.

en-ru

Source: A few years ago, I called for toothpaste from my hotel room, and it was delivered to me by a robot.

TransLLaMa: Несколько лет назад я попросила зубную пасту из моего номера в отеле и она была доставлена мне роботом.

Back translation: A few years ago I asked for toothpaste from my hotel room and it was delivered to me by robot.

Source: It conducted the first quantum video call, and it launched the first drone that can carry a passenger in air.

TransLLaMa: Она провела первую видео-конференцию с использованием квантовых технологий и запустила первый беспилотный самолет, способный перевозить пассажира в воздухе.

Back translation: She held the first video conference using quantum technology and launched the first unmanned aircraft capable of carrying a passenger in the air.

Source: And it's really gratifying to hear from people like Raymond who are using the technology we are building, and others, for so many wonderful things.

TransLLaMa: И это действительно приятно слышать от людей, как Раймонд, которые используют технологии, которые мы создаём, и другие, для таких замечательных вещей.

