# OpenReview forum: "TransLLaMa: LLM-based Simultaneous Translation System"
_ICLR.cc/2024/Conference — Submitted to ICLR 2024_

### Official Review · Reviewer_yShq · 2023-10-28

**Soundness:** 3 good
**Presentation:** 3 good
**Contribution:** 3 good
**Rating:** 8
**Confidence:** 5

**Summary:**

The paper explore the use of LLMs in the tasks of simultaneous translation (SiMT), which means "Translate as we speak". The task involves training a policy that, given a partial input, decides whether to continue to listen for more words from speaker, or to translate right away with the partial input. The goal is to achieve optimally low latency and high accuracy.
The paper does so with LLMs by finetuning the model on a causal alignment data for such task, which given a partial input, decides to generate the translation or a <wait> token to collect more context for the partial input.
The results show comparable with existing SiMT baselines.

**Strengths:**

- Though there are many papers about training LLMs to as decision-making agent, I consider doing so with Simultaneous translation, which is predominantly about speech, is novel and the task of SiMT can improved with the help from LLM.
- The results show comparable with existing high-quality SiMT baselines, though I highly doubt the actual computational cost is anywhere comparable (see weakness). Future work should make up for this by achieve higher translation quality and latency, as well as in other lower-resource languages.

**Weaknesses:**

- Repeated inference of LLM is a huge computational cost, everytime a <wait> token is generated, the text prompt is updated and many tokens representations have to be recalculated without a theoretical room for caching. As such, real-life inference, with a fixed physical hardware, will be much slower compared to existing lightweight translation model. This is true for closed-source GPT models as well, as more tokens called to the API leads to more expensive bill.
   - Therefore, I urge the authors to provide a real-life inference cost/wall-time comparison to have a better picture of the cost trade-off here and makes the paper complete. I would appreciate and change scores if such report is produced.

**Questions:**

- There are many papers about using LLMs as decision-making agent, please cite them.

---

> ### Author Response · Authors · 2023-11-13
> **Response to Reviewer yShq**
>
> ***Repeated inference of LLM is a huge computational cost ... real-life inference, with a fixed physical hardware, will be much slower compared to existing lightweight translation model. This is true for closed-source GPT models as well, as more tokens called to the API leads to more expensive bill.***
>
> It is true that our model performs a forward pass on the entire prompt each time it is updated with both the incremental partial input and translation. It is also true that the compute cost of running an off-the-shelf LLM – even a relatively small one (e. g. 13 billion parameters) – is much higher than that of the latest published SiMT models. State-of-the-art SiMT systems typically rely on relatively small encoder-decoder transformers, such as mBART (~690M parameters), giving them an undeniable speed advantage at inference over even a modestly sized LLM (such as LLaMa 13B). We believe, however, that inference costs will likely become less of a bottleneck with faster existing (such as H100) and future hardware, as well as more efficient quantization techniques. Also, competition in the closed-source space is likely to drive down the costs of proprietary LLMs even further. Regardless of the cost, we believe that the potential of LLMs for SiMT is far from exhausted. The fact that translation with GPT-4 (see Appendix E) – which is presumably an order of magnitude larger than Llama 70B – is only about 50% slower than real-time in T2TT mode, suggests that with efficient implementation optimized for fast inference it should be possible to achieve real-time or faster-than-real-time translation for smaller models (such as Llama-2).
>
> ***...I urge the authors to provide a real-life inference cost/wall-time comparison to have a better picture of the cost trade-off here and makes the paper complete. I would appreciate and change scores if such report is produced.***
>
> We have compared our method to the other state-of the-art model on inference wall time and summarize the results in Table 4 in Appendix E. In general, inference times are much longer for our model than for the smaller models we used as baselines (in Fukuda et al. 2023 and Papi et al, 2023). While our model is admittedly slower, several things could be done in future work to improve inference speed:
>
> (1)	Use a shorter system message in the prompt or remove the prompt altogether. The system message we used in the paper is much longer than the average length of the sentence. In fact, the system message is only necessary if the base model is used zero-shot (that is without prior supervised fine-tuning). This can dramatically (by about ~40%) shorten the inference time.
>
> (2)	Use more efficient quantization schemes. We noticed that loading the model in 4 bit increases inference time by about 43% compared to the native 16-bit quantization. We speculate that with more efficient quantization techniques and low bit width arithmetic hardware support (this is a currently very active area of research), inference speed is likely to become less of an issue from a practical point of view.
>
> (3)	Take an end-to-end approach (as in [1, 2]), in which input audio is directly embedded (e. g. with Wav2Vec or HuBERT), reducing delays due to a separate ASR subsystem. In fact, some recent works [3] have already showed that raw audio chunks can be embedded and (after projecting with an adapter) be injected into the prompt as if they were text embeddings.
>
> Encouragingly, GPT-4 (despite being presumably at least an order of magnitude bigger) performs much faster than Llama on the same task. This suggests that even very large language models can, if served efficiently and with proper optimizations, become a viable alternative to current compact SiMT systems. Finally, since Lllama2 (as many other LLMs) are trained predominantly on English text, the tokenizer encodes English more efficiently (that is with fewer tokens on average) than other, less represented, languages. For example, we made three texts from sentences sourced from TED talks, each 11500 characters long, in English, German and Russian and encoded them with the Llama-2 tokenizer. The resulting numbers of tokens were 2890, 3397 and 4129, respectively. This suggests that translating into English will be faster than from English. We now mention these considerations in the final paragraph of the last section (lines 252-265).
>
> [1] Fukuda et al. (2023) NAIST simultaneous speech-to-speech translation system for IWSLT 2023. doi: 10.18653/v1/2023.iwslt-1.31.
>
> [2] Papi et al. (2023) Attention as a guide for simultaneous speech translation. In Proceedings of the 61st Annual Meeting of the ACL.
>
> [3] Fathullah et al. (2023). Prompting Large Language Models with Speech Recognition Abilities. ArXiv.
>
> ***There are many papers about using LLMs as decision-making agent, please cite them.***
>
> Thank you very much for this suggestion. We now mention this and cite some recent papers in the introduction section (L41-42 of the updated manuscript).

---

### Official Review · Reviewer_Hphj · 2023-10-30

**Soundness:** 3 good
**Presentation:** 3 good
**Contribution:** 2 fair
**Rating:** 6
**Confidence:** 4

**Summary:**

This work investigates the use of LLMs for simultaneous MT (SiMT). The training data is preprocessed by generating word alignments, and then inserting enough <WAIT> tokens into the target sequence such that no alignment link has a higher source than target position (called "casually aligned dataset"). A variation of the commonly used wait-k strategy is used for inference.

**Strengths:**

The setup is described clearly and is very straightforward, which makes this work easily reproducible. I also appreciate the results section, which includes the most natural ablations and is not overselling the results. In fact, the most obvious concern of using LLMs for SiMT - inference time - is acknowledged in the paper. The evaluation is based on (just) two language pairs and two LLM (sizes), which is definitely on the slim side, but it meets the minimum bar for me.

**Weaknesses:**

I don't think that this paper is particularly innovative. On a high level, it strikes me as one of the "we tried LLMs for task X and it worked" papers that are very common these days. That being said, I think that this is one of the better papers in that category due to the sober evaluation and clear writing. So although I wasn't inspired by this work, there is still value in publishing it for the sake of completeness of the body of literature on LLMs.

Fig. 2 looks broken.. I guess the key point here is that "away" is aligned to "befreite", but the alignment link is not shown in the original en-de alignment.

**Questions:**

- Have you compared LoRA fine-tuning with prompt-tuning or full fine-tuning (at least with the small LLMs)?
- Have you tried small (<=1B) LLMs that would be more practical in a real-life SiMT scenatio?

---

> ### Author Response · Authors · 2023-11-13
> **Response to Reviewer Hphj**
>
> ***Fig. 2 looks broken.. I guess the key point here is that "away" is aligned to "befreite", but the alignment link is not shown in the original en-de alignment.***
>
> Thank you for the opportunity to clarify this. The links are established by SimAlign (as we mention on line 123 in the revised manuscript). Indeed, the source word “away” would be aligned with “befreite”, but SimAlign does not treat phrasal verbs (such as “took away” in this case) as a whole, but rather as separate words. In this example, the constituent words of that verb (“took” and “away”) could not be directly linked to any German worlds in the target. This explains the missing link. As we suggest in the paper (lines 116-120 in the revised manuscript), the alignment does not need to be perfect for our method to work.
>
> ***Have you compared LoRA fine-tuning with prompt-tuning or full fine-tuning (at least with the small LLMs)?***
>
> No, we did not explore other fine-tuning techniques. While full fine-tuning would likely slightly enhance the translation quality (as it generally does in other use cases), it would require significantly more effort and time. Specifically, fine tuning the 70B Llama2 would require splitting the model over more than one device (using LoRA and 4-bit quantized weights allowed us to find-tune Llama2-70B on one device (A100 80GB)). Even with the smaller Llama-13B, full fine-tuning would only be possible with in the native 16-bit quantization, and most likely (although we did not try) require a complicated multi-GPU setup and a long time.
>
> ***Have you tried small (<=1B) LLMs that would be more practical in a real-life SiMT scenario?***
>
> No, we have not. While a small LLM would run faster (and therefore be more practical in the real world), our focus was on exploring the utility and potential of a decoder-only transformer for the SiMT task. These decoder-only transformers tend to come in sizes much larger than 1B parameters. For example, the smallest version of Llama2, the latest open-source LLM at the time of writing the paper, has 7B parameters. There are of course older decoder-only transformers (e. g. GPT-2), but given that already at 13B parameters Llama2 performed significantly worse than at 70B parameters, we speculate that much smaller models would fail the SiMT task unless fully trained on a very large dataset of source-target pairs.

---

> ### Comment · Area_Chair_XxEj · 2023-11-20
>
> Hello, reviewer. Please review the author's response to see if it resolves your concerns.

---

### Official Review · Reviewer_jikd · 2023-11-01

**Soundness:** 4 excellent
**Presentation:** 3 good
**Contribution:** 3 good
**Rating:** 6
**Confidence:** 4

**Summary:**

This study focuses on simultaneous machine translation (SiMT). It explores the use of decoder-based language models for this purpose. The experiments conducted on English-German and English-Russian translations show results that are comparable to those of current state-of-the-art baselines.

**Strengths:**

- The concept of employing large language models for simultaneous translation appears both novel and exciting.
- The paper is clearly written and easy to understand.
- The related work section provides a comprehensive summary of simultaneous translation research in the field of natural language processing.

**Weaknesses:**

- Figure 1 lacks clarity in terms of distinguishing when specific actions (READ/WRITE) occur. It would be more reader-friendly if the figure illustrated a step-by-step walkthrough (e.g., t=1, t=2, t=3).
- In simultaneous translation, wall-clock time (actual speed) is a critical factor. It would be important to report or at least mention how long it takes to generate translations in this setting.
- The experiment only presents BLEU scores; it lacks concrete examples of output, which would be beneficial for understanding the translation quality.

**Questions:**

- I wonder if there are any experimental results for a setting when the target language is English such as DE-EN (instead of EN-DE). Since LLMs are typically trained mainly in English, I wonder if it makes a difference in the performance between En-X and X-En.
- Line 161. "We did not use beam search during generation." Is there any reason not to use beam search?

---

> ### Author Response · Authors · 2023-11-13
> **Response to Reviewer jikd**
>
> ***Figure 1 lacks clarity in terms of distinguishing when specific actions (READ/WRITE) occur. It would be more reader-friendly if the figure illustrated a step-by-step walkthrough (e.g., t=1, t=2, t=3).***
>
> Thank you for this suggestion. We hope that the updated figure is much clearer now.
>
> ***In simultaneous translation, wall-clock time (actual speed) is a critical factor. It would be important to report or at least mention how long it takes to generate translations in this setting.***
>
> Thank you for raising this important point. We have added a section (Appendix E) offering a comprehensive comparison of generation times (relative to the duration of input audio) for our approach and selected baselines. We report real-time factors in different settings providing insights about delays introduced by ASR, base model quantization and size, as well as the presence of the system message. We hope the Reviewer will find this additional information satisfactory.
>
> ***The experiment only presents BLEU scores; it lacks concrete examples of output, which would be beneficial for understanding the translation quality.***
>
> Thank you for this suggestion. We have added some examples to Appendix F of the revised manuscript.
>
> ***I wonder if there are any experimental results for a setting when the target language is English such as DE-EN (instead of EN-DE). Since LLMs are typically trained mainly in English, I wonder if it makes a difference in the performance between En-X and X-En.***
>
> We report results only for translation from English because we utilized MuST-c, a common and easily available dataset used to train and benchmark SiMT systems, which includes talks in English and their translations into another language. However, because the base LLM is pre-trained mostly on English text, we can speculate that translation into English should be better that into a foreign language. Moreover, translation into English should be faster, because fewer tokens on average are needed to encode an English text than a text of the same size (in terms of the number of characters) in another, less represented language (e. g. three texts made from sentences sourced from TED talks, each 11500 characters long, in English, German and Russian are encoded into 2890, 3397 and 4129 tokens, respectively, by the Llama-2 tokenizer. We now mention this in the final paragraph of the last section (lines 252-265).
>
> ***Line 161. "We did not use beam search during generation." Is there any reason not to use beam search?***
>
> Thank you for this question. Although we found beam search to improve translation quality (in locally run Llama2 models), it is not clear how using beam search would affect GPT-3.5 and GPT-4 translation performance. This is because OpenAI API does not expose any beam search-related settings (although it might use them internally). Also, beam search slows down the generation process. For these two reasons, we chose to focus on greedy search.

---

> ### Comment · Area_Chair_XxEj · 2023-11-20
>
> Hello, reviewer. Please review the author's response to see if it resolves your concerns.

---

> > ### Comment · Reviewer_jikd · 2023-11-23
> >
> > Thank you for your response. I now have a better understanding of the concept and will raise my score accordingly. However, considering the substantial amount of new information received during the discussion period, I believe the paper could further improve in terms of clarity, especially in its wording and table format.

---

> > > ### Author Response · Authors · 2023-11-23
> > >
> > > We sincerely thank the Reviewer for the positive feedback about our paper. In addition to the improvements we made in the latest revision of the paper, we have added additional text in Appendix C to clarify Tables 2 and 3.

---

### Official Review · Reviewer_xrvK · 2023-11-02

**Soundness:** 2 fair
**Presentation:** 2 fair
**Contribution:** 3 good
**Rating:** 5
**Confidence:** 4

**Summary:**

This paper presents a method to enhance the performance of Language Model-based Simultaneous Machine Translation (SiMT) systems. The authors propose fine-tuning a pre-trained Language Model (LLM) using direct supervision on a dataset of causally aligned source-target sentence pairs. They demonstrate that the LLM can achieve simultaneous translation and input segmentation without the need for a separate policy, with performance that matches or surpasses existing state-of-the-art systems. The paper provides an overview of recent SiMT literature, details the system's architecture, data preparation, and training procedure, and showcases its performance on different language directions. The authors also discuss the limitations of their approach and suggest future research directions. Overall, the paper contributes a novel approach to improving SiMT systems by leveraging fine-tuning of pre-trained LLMs.

**Strengths:**

The paper introduces a novel approach to improving SiMT systems by fine-tuning a pre-trained Language Model (LLM) with direct supervision on causally aligned source-target sentence pairs. This approach differs from previous methods that rely on separate policies or incremental decoding. By leveraging the capabilities of LLMs, the paper offers a fresh perspective on enhancing SiMT performance.

**Weaknesses:**

One of the main concerns regarding this paper is the reliance on a reference-based approach for the causal alignment introduced. While the paper claims to propose a novel method, similar ideas have been studied in previous simultaneous translation literature (e.g. [1]). However, the paper lacks a comparative analysis with these existing approaches in the experiment section, making it difficult to assess the novelty and superiority of the proposed method.

Furthermore, a significant limitation of the reference-based approach is the potential mismatch between full sentence translation and simultaneous translation. The references used to generate the causal alignment are derived from complete sentence translations, which may not be suitable for the dynamic nature of simultaneous translation. Simultaneous translation requires the model to begin translation based on partial context, and the reference-based approach may not adequately capture the challenges and nuances specific to this task.

[1] Simultaneous translation policies: from fixed to adaptive. ACL, 2020

**Questions:**

1. Is the comparison in Figure 6 fair, considering that the authors only evaluate their method and two state-of-the-art models based on translation quality without considering latency?

2. Did the authors observe a high variance in the latency of the resulting causal alignment, potentially due to the fact that the gold reference used is designed for full sentence translation rather than simultaneous translation?

---

> ### Author Response · Authors · 2023-11-13
> **Response to Reviewer xrvK**
>
> ***While the paper claims to propose a novel method, similar ideas have been studied in previous simultaneous translation literature (e.g. [1]). However, the paper lacks a comparative analysis with these existing approaches in the experiment section, making it difficult to assess the novelty and superiority of the proposed method.***
>
> Without claiming superiority of our approach (in its current implementation) in either computational cost or quality, we aimed to demonstrate, for the first time, that a general-purpose (that is, not pre-trained specifically for translation) decoder-only LLM can perform the SiMT task. The potential of such LLMs for SiMT can be unlocked, as we demonstrate, with minimal fine-tuning. The fact that causal alignment does not have to be perfect for the method to work suggests that further improvements can be achieved with a refined  alignment procedure and/or human simultaneous translations used as targets. We do make an effort to compare our model to a selection of published state-of-the-art SiMT systems. Additionally, we compare these systems and ours in terms of wall time it takes to process one second of audio (Appendix E in the revised manuscript).
>
> ***One of the main concerns regarding this paper is the reliance on a reference-based approach for the causal alignment introduced… a significant limitation of the reference-based approach is the potential mismatch between full sentence translation and simultaneous translation. The references used to generate the causal alignment are derived from complete sentence translations, which may not be suitable for the dynamic nature of simultaneous translation. Simultaneous translation requires the model to begin translation based on partial context, and the reference-based approach may not adequately capture the challenges and nuances specific to this task.***
>
> Thank you for raising this important point. Despite being standard practice in the SiMT literature, using full sentence translations as references for causal alignment has its limitations and may not always be optimal. Indeed, offline translations quite often violate causality (later source words might appear earlier in the target). Causality violation happens less often in simultaneous translation, because interpreters are under pressure to produce translation as early as possible to avoid working memory overload (although causality can often be violated when interpreters translate what has not yet been said (e.g. [1]). On the whole, we agree that offline and simultaneous translations are quite different things, with the latter potentially more suitable for the purpose of causal alignment and LLM fine-tuning. However, due to difficulty of obtaining high-quality datasets of simultaneous translations, we chose to follow the standard practice in the SiMT literature of using TED talks, many of which are translated offline into languages other than English. Sticking to TED talks makes comparisons with previous approaches easier (e. g. SiMT models submitted to IWSLT in the past few years have been tested on the MuST-c dataset primarily sourced from TED talks). Importantly, as we emphasize in our paper, our approach works with encouraging results despite the imperfect alignment procedure and offline targets.
>
> [1] Amos et al. (2022), Prediction during simultaneous interpreting: Evidence from the visual-world paradigm. Cognition.
>
> ***Is the comparison in Figure 6 fair, considering that the authors only evaluate their method and two state-of-the-art models based on translation quality without considering latency?***
>
> Thank you for this question. The comparison is made as fair as possible by keeping average lagging (AL) below about 2000 ms (which we briefly mention on lines 201-202). We also do provide comparative latency performance in Appendix C (Table 3, p. 15). To help the reader, we now refer the reader to Table 3 it in the caption of Figure 6.
>
> ***Did the authors observe a high variance in the latency of the resulting causal alignment, potentially due to the fact that the gold reference used is designed for full sentence translation rather than simultaneous translation?***
>
> Thank you for the question. The average translation latency (measured by SimulEval) might depend on the average source-target “gap” (measured in words) resulting from causal alignment, which shifts target words into the future. Since for German, this “gap” is generally larger than for Russian, it is possible for latency metrics for English-German to be larger than those for English-Russian translation. However, based on Table 2 and 3, we did not observe a striking latency difference between these language pairs. Nor did we observe large differences in the variance of the latency estimates between the language pairs tested. A different picture might emerge if causal alignment were performed on simultaneous (rather than offline) translation targets. We leave this to future work.

---

> > ### Comment · Area_Chair_XxEj · 2023-11-20
> >
> > Hello, reviewer. Please review the author's response to see if it resolves your concerns.

---

### Author Response · Authors · 2023-11-13
**Note to all Reviewers**

We sincerely thank all the Reviewers for their thoughtful comments and suggestions, as well as the time they took to carefully review our paper. We have worked hard to address all the concerns and questions in as much detail as possible and sincerely hope that the Reviewers will find our responses satisfactory. In addition to our responses, we added two new appendices (E and F), improved Figure 1, and made other changes to the manuscript (we specify the places in which the changes were made).

---

### Meta-Review · Area_Chair_XxEj · 2023-12-03

**Metareview:**

The paper presents a novel application of pre-trained LLMs in the field of SiMT, demonstrating their potential with minimal fine-tuning. Supported by experiments in English-German and English-Russian translations, this approach is effective, with results comparable to current benchmarks. The initial omission of translation speed metrics was a concern, as speed is critical in SiMT settings. But this was later addressed during the rebuttal.

While the paper presents a promising approach to SiMT using LLMs, it faces challenges in demonstrating the novelty, as similar ideas have been previously explored. The paper's contribution is to show that the existing methods of SiMT can be used in LLMs.

**Justification For Why Not Higher Score:**

The paper's inability to convincingly demonstrate novelty in its approach is a key factor behind the rejection. While the paper successfully applies pre-trained LLMs to SiMT, the concept and methods it employs have been previously explored in the field. This overlap with existing literature diminishes the perceived originality and contribution of the paper.

**Justification For Why Not Lower Score:**

N/A

---

### Decision · Program_Chairs · 2024-01-16

Reject